# MRI Relaxometry for Quantitative Analysis of USPIO Uptake in Cerebral Small Vessel Disease

**DOI:** 10.3390/ijms20030776

**Published:** 2019-02-12

**Authors:** Michael J. Thrippleton, Gordon W. Blair, Maria C. Valdes-Hernandez, Andreas Glatz, Scott I. K. Semple, Fergus Doubal, Alex Vesey, Ian Marshall, David E. Newby, Joanna M. Wardlaw

**Affiliations:** 1Centre for Clinical Brain Sciences, University of Edinburgh, Edinburgh EH16 4SB, UK; m.j.thrippleton@ed.ac.uk (M.J.T.); gordon.blair@ed.ac.uk (G.W.B.); M.Valdes-Hernan@ed.ac.uk (M.C.V.-H.); andi@mingsze.com (A.G.); fergus.doubal@ed.ac.uk (F.D.); ian.marshall@ed.ac.uk (I.M.); 2Edinburgh Imaging, University of Edinburgh, Edinburgh EH16 4TJ, UK; scott.semple@ed.ac.uk (S.I.K.S.); d.e.newby@ed.ac.uk (D.E.N.); 3UK Dementia Research Institute at the University of Edinburgh, Edinburgh EH16 4SB, UK; 4Centre for Cardiovascular Sciences, University of Edinburgh, Edinburgh EH16 4TJ, UK; alexvesey@gmail.com

**Keywords:** cerebral small vessel disease, MRI, ferumoxytol, USPIO, inflammation, relaxometry

## Abstract

A protocol for evaluating ultrasmall superparamagnetic particles of iron oxide (USPIO) uptake and elimination in cerebral small vessel disease patients was developed and piloted. *B*_1_-insensitive *R*_1_ measurement was evaluated in vitro. Twelve participants with history of minor stroke were scanned at 3-T MRI including structural imaging, and *R*_1_ and *R*_2_* mapping. Participants were scanned (i) before and (ii) after USPIO (ferumoxytol) infusion, and again at (iii) 24–30 h and (iv) one month. Absolute and blood-normalised changes in *R*_1_ and *R*_2_* were measured in white matter (WM), deep grey matter (GM), white matter hyperintensity (WMH) and stroke lesion regions. *R*_1_ measurements were accurate across a wide range of values. *R*_1_ (*p* < 0.05) and *R*_2_* (*p* < 0.01) mapping detected increases in relaxation rate in all tissues immediately post-USPIO and at 24–30 h. *R*_2_* returned to baseline at one month. Blood-normalised *R*_1_ and *R*_2_* changes post-infusion and at 24–30 h were similar, and were greater in GM versus WM (*p* < 0.001). Narrower distributions were seen with *R*_2_* than for *R*_1_ mapping. *R*_1_ and *R*_2_* changes were correlated at 24–30 h (*p* < 0.01). MRI relaxometry permits quantitative evaluation of USPIO uptake; *R*_2_* appears to be more sensitive to USPIO than *R*_1_. Our data are explained by intravascular uptake alone, yielding estimates of cerebral blood volume, and did not support parenchymal uptake. Ferumoxytol appears to be eliminated at 1 month. The approach should be valuable in future studies to quantify both blood-pool USPIO and parenchymal uptake associated with inflammatory cells or blood-brain barrier leak.

## 1. Introduction

Cerebral small vessel disease (SVD) accounts for 20–25% of strokes and increases the risk of cognitive impairment, disability and dementia. The pathogenesis is poorly understood but several mechanisms are hypothesised to contribute, including inflammation [1], blood-brain barrier (BBB) leakage [2] and vessel stiffness [3]. Several magnetic resonance imaging (MRI) methods have been used to explore microstructural and microvascular properties in SVD, including those utilising intrinsic tissue contrast (e.g., diffusion tensor imaging to assess white matter integrity) [4], respiratory gas challenges (e.g., cerebrovascular reactivity) [5], and gadolinium contrast-based approaches (e.g., dynamic contrast-enhanced (DCE) MRI to quantify BBB permeability) [6]. As far as we are aware, no studies have employed ultrasmall superparamagnetic particles of iron oxide (USPIO) contrast agents to study SVD. These particles, which have *T*_1_, *T*_2_ and *T*_2_* shortening effects, have several benefits compared with gadolinium-based contrast agents: (i) a large *T*_2_* effect due to the superparamagnetic nature of the contrast agent, (ii) they may often be used in patients with renal failure, for whom gadolinium-based agents are contraindicated [7], (iii) slow rates of extravasation and elimination facilitate convenient MR angiography at high resolution and steady-state cerebral blood volume (CBV) measurement [8], and (iv) phagocytosis by macrophages, which concentrate in inflamed tissue, provides an imaging marker of inflammation [9], for example in abdominal aortic aneurysm [10] and myocardial infarction [11]. However, since inflammatory uptake typically occurs over a longer period than the MRI examination, multiple examinations are required and quantitative analysis based on signal changes is therefore impractical. 

In this pilot study, we developed and evaluated an MRI relaxometry protocol for assessing USPIO (ferumoxytol) uptake in patients with cerebral SVD. We measured *R*_2_* using the three dimensional multiple-spoiled-gradient-echo technique. Volumetric mapping of *R*_1_ within a feasible scan time is challenging at 3T, where increased *B*_1_ inhomogeneity impairs the accuracy of the widely-used and widely-available variable flip angle method; we therefore adapted the DESPOT1-HIFI approach proposed by Deoni et al. so that small changes in *R*_1_ obtained across different examinations could be measured [12]. We first scanned a multi-compartment test object containing a range of *R*_1_ values to confirm the accuracy of *R*_1_ mapping. We then studied patients following recent minor ischaemic stroke, since SVD features are prevalent in such patients compared with the wider population [13,14]. Specifically, we aimed to (i) validate *B*_1_-insensitive *R*_1_ measurement against a reference-standard technique, (ii) determine whether *R*_1_ and *R*_2_* changes in the brain are detectable following USPIO infusion and at one day following infusion, (iii) determine whether residual USPIO is detectable one month after administration by obtaining an additional *R*_2_* measurement (*R*_1_ was not measured at this time point in order to reduce the burden on participants), and (iv) measure blood-normalised *R*_1_ and *R*_2_* changes in grey matter (GM), normal-appearing white matter (WM) and white matter hyperintensity (WMH) tissue regions in this patient group in order to estimate cerebral blood volume and test for potential extravascular or perivascular USPIO uptake due to inflammation or BBB leak. Blood *R*_1_ was measured in the superior sagittal sinus and used for normalisation of both *R*_1_ and *R*_2_* changes in tissue, since *R*_2_* changes in blood were spatially heterogeneous and inconsistent across patients. 

## 2. Results

### 2.1. Compliance, Tolerability and Symptoms

*R*_1_ and *R*_2_* maps, and structural images were obtained successfully in all 12 patients (mean age 74.7 ± 6.3 years; Table 1). Scan 2 occurred on average 28 h (range: 26–30 h) after scan 1_pre_ for all patients, while scan 3 took place an average of 31 days (range: 22–36 days) after scan 1_pre_. Values of *R*_1_ and Δ*R*_1_ for one patient at scan 1_post_ were outliers and were excluded from analyses; no signs of protocol deviations or of unusual observations in relation to this participant were found in the study records or image header information.

### 2.2. In Vitro Accuracy of R_1_ Mapping

*R*_1_ measurements in a MRI phantom (test object) obtained using the variable flip angle approach were inaccurate compared with gold-standard inversion-recovery spin-echo measurements (absolute difference: 0.03–0.58 s^−1^; Figure 1a). The errors were largely explained by the spatial flip angle variation (measured using DESPOT1-HIFI), which was found to be greater than the nominal value in the five regions with lowest *R*_1_ and vice versa. DESPOT1-HIFI *R*_1_ measurements corresponded more closely to gold-standard measurements across the full range of values provided the equation of Deichmann et al. [15] was used to model the IR-sGRE signal (absolute difference: 0.01–0.23 s^−1^; Figure 1a). Results obtained using a simplified IR-sGRE signal model (absolute difference: 0.01–0.87 s^−1^; Figure 1a) were acceptable within the range of *R*_1_ found in GM and WM but were inaccurate at the highest *R*_1_ value, which is typical of blood magnetisation following USPIO administration. To explore this further, the signals predicted by both models and by the full mathematical description by Brix et al. were calculated as a function of *R*_1_ (Figure 1b), demonstrating substantial errors in the simplified model predictions at high *R*_1_ values. Patients’ measurements were therefore obtained using the DESPOT1-HIFI method with the Deichmann signal model.

### 2.3. Relaxation Rate Changes Following USPIO Administration

Example parametric maps are shown in Figure 2, illustrating the subtle *R*_1_ and *R*_2_* changes observed in patients. *R*_1_ increased relative to baseline in all regions at scan 1_post_ and scan 2 (Table 2, Figure 3a; *p* < 0.05). *R*_2_* also increased in all regions at scan 1_post_ and scan 2 (Table 2, Figure 3b; *p* < 0.01). There was a trend towards increasing *R*_1_ between scan 1_post_ and scan 2 in all regions, but this was statistically significant only in blood (*p* < 0.01) and WMH (*p* < 0.05); for *R*_2_*, a significant further increase between scan 1_post_ and scan 2 was seen in WM, GM, WMH (*p* < 0.001) and in stroke lesions (*p* < 0.05). Relaxation rates corresponding to each subject, region and time point are provided as Appendix A.

The mean change in *R*_2_* at scan 3 (one month) was small (0.006–0.327 s^−1^) and was not statistically significant in any of the regions. There was no association between changes in the two relaxation rates Δ*R*_1_ and Δ*R*_2_* immediately following USPIO administration except in WMH (*p* < 0.05), but there was a strong association in all regions at 24–30 h (*p* < 0.01; Table 3, Figure 4).

### 2.4. Blood-Normalised Relaxation Rate Changes

The blood-normalised relaxation rate changes ∆*R*_1__,norm_ and $\Delta R_{2,\mathrm{norm}}^{*}$ (Table 4, Figure 5) did not change between scan 1_post_ and scan 2 in any of the tissue regions—in other words, the tissue relaxation rate changes over this period are accounted for by the changes in capillary USPIO concentration following infusion of the final two-thirds dose, and elimination and redistribution of the contrast agent. Values were greater in GM than in WM for both measures and time points (*p* < 0.001). ∆*R*_1__,norm_ at scan 2 was also greater in WMH and SL versus WM (*p* < 0.05); there were no other significant differences between tissues. In general, a narrower distribution of values (expressed as the coefficient of variation; Table 4) was seen for $\Delta R_{2,\mathrm{norm}}^{*}$ compared with ∆*R*_1__,norm_.

## 3. Discussion

In this study we administered USPIO and performed *R*_1_ and *R*_2_* mapping in patients with SVD. The USPIO were tolerated, and importantly, we found no evidence of retention at one month. We established a protocol for assessing contrast uptake, including *R*_1_ mapping in the presence of the substantial *B*_1_ inhomogeneity seen at 3T and validated this in vitro. Using relaxometry, we were able to detect changes in both normal and diseased tissues immediately after USPIO administration at one third of the standard dose, and at one day following administration of the full dose. Blood-normalised changes in the relaxation rates in WM and GM regions were consistent with expected differences in cerebral blood volume, and did not change further at 24–30 h, which implies that no detectable parenchymal uptake occurred on this timescale. Changes in *R*_1_ and *R*_2_* were strongly correlated at 24–30 h; the weak association immediately post-contrast is likely due to fewer subjects receiving scan 1_post_ (*n* = 8). Narrower distributions were seen for *R*_2_*-compared with *R*_1_-derived quantities, suggesting that the former may be a more sensitive marker of uptake.

Few previous studies have employed quantitative MRI to assess USPIO uptake in the brain, and we are unaware of any such application in cerebral small vessel disease patients. A small number of studies have administered USPIO in stroke patients, with observations of signal change, mainly on *T*_1_w images, that can be cautiously ascribed to inflammatory uptake in some stroke lesions within 1–2 weeks following acute ischaemic stroke [16,17,18,19]. A key advantage of using quantitative relaxometry, as here, is that uptake occurring over a longer time period can be measured; approaches based on changes in signal intensity are restricted to qualitative interpretation or to measuring uptake during the examination (as in DCE and dynamic susceptibility contrast MRI). By performing quantitative relaxometry including blood pool measurements, we were also able to determine whether the observed changes could be explained by intra-vascular (as opposed to parenchymal) USPIO uptake. 

The approach presented in this paper relies on accurate volumetric measurement of *R*_1_ and *R*_2_*. Through selection of an appropriate acquisition strategy and careful signal modelling, we were able to measure inter-examination *R*_1_ changes, obtaining GM, WM and venous blood *R*_1_ values similar to those reported in the literature [20,21,22]. While the alternative variable flip angle (VFA) method is faster than DESPOT1-HIFI (since the additional IR-sGRE scans are relatively time-consuming), the former approach yields parameter maps that are strongly affected by spatial variation in the flip angle, which can be substantial at 3T and/or when slab-selective excitation pulses are used. Our phantom measurements confirmed the accuracy of DESPOT1-HIFI across a range of *R*_1_ provided appropriate signal models are used. Alternatively, it is possible to reduce the scan time by combining VFA acquisition with a technique that directly maps the flip angle, such as actual flip angle imaging (AFI) [23], but reliable flip angle mapping sequences are not widely available on commercial MRI scanners. *R*_2_* values are necessarily protocol-dependent and are affected by macroscopic *B*_0_ field inhomogeneities, and there is no accepted gold-standard method. However, the values measured in thalamus at baseline were similar to those reported previously in volunteers [24,25].

As expected, both *R*_1_ and *R*_2_* increased following USPIO administration. At one-month post-infusion, *R*_2_* had returned to baseline with no statistically significant changes detected. The average one-month changes in *R*_2_* corresponded to 0.2% (WM), 3.4% (GM), 1.2% (WMH) and 6.7% (SL) of the average changes measured at 24–30 h, implying elimination or near-elimination of ferumoxytol from the regions measured on this timescale.

Although *R*_1_ and *R*_2_* increases were detected in the brain tissue of all patients following USPIO administration, such measurements do not distinguish between USPIO in the blood pool from uptake in parenchyma. We therefore obtained additional measurements immediately after infusing a third of the full dose, at which point it is reasonably assumed that any uptake is intravascular and that the normalised relaxation rate changes, ∆*R*_1__,norm_ and $\Delta R_{2,\mathrm{norm}}^{*}$, are respectively equivalent or proportional to the cerebral blood volume fraction. ∆*R*_1__,norm_ measurements were consistent with plasma volume fraction (*v*_p_) measurements obtained in a similar patient group by DCE-MRI [6], and were greater in GM than in WM as expected. A similar pattern was seen for $\Delta R_{2,\mathrm{norm}}^{*}$ measurements but there was no significant difference between WM and WMH values. Varallyay et al. used changes in *T*_2_*w signal intensity to assess relative CBV in patients with a variety of brain pathologies [8]. The approach described here extends this by measuring both *R*_1_ and *R*_2_*, and by normalising for blood-pool USPIO concentration, thereby enabling between as well as within-subject comparison.

Neither normalised relaxation rate changed significantly at 24–30 h following USPIO infusion. This suggests that there was no detectable parenchymal uptake—for example, due to accumulation of inflammatory cells or BBB leak—and that the signal changes observed are explained by the blood-pool uptake alone. In the event of parenchymal uptake occurring, there would likely be increased ∆*R*_1__,norm_ and $\Delta R_{2,\mathrm{norm}}^{*}$ at 24–30 h. However, it is important to note that our sample size was small and that any generalised chronic neuroinflammation in SVD is likely to be subtle compared with that found in some diseases of the large arteries. Larger studies, ideally with histological or biochemical validation, are required to draw biological conclusions. Finally, we observed a strong linear association between, ∆*R*_1__,norm_ and $\Delta R_{2}^{*}$ at 24–30 h, suggesting that either quantity can be used to assess uptake.

Advantages of our work include the use of two quantitative relaxometry methods, additional post-contrast measurement to differentiate blood-pool and parenchymal USPIO uptake, and a final measurement to confirm elimination of ferumoxytol from the brain. Our *R*_1_ mapping protocol was validated in vitro and is resistant to *B*_1_ inhomogeneity. In contrast to acute ischaemic stroke, where inflammation may be directly visualised in the acute and sub-acute phases, chronic inflammation in SVD is likely to be more subtle and diffuse. It is therefore important to establish sensitive methods such as these for detecting changes that may not be radiologically visible. We recruited patients with past minor stroke including lacunar stroke since these subjects typically have a higher burden of small vessel disease features and a small index infarct, which can easily be separated from the other tissues of interest. Patients were scanned weeks or months following the index stroke, which also minimised the effect of any acute stroke changes on the background brain condition. The main limitation of this pilot study is the sample size, which precludes firm biological inferences.

Our data suggest that future studies of SVD (and other pathologies) using USPIO contrast agents would benefit from obtaining quantitative relaxation rate maps including immediately post-infusion to differentiate intravascular and parenchymal uptake. With regard to *R*_1_ versus *R*_2_*-based approaches, there are theoretical grounds for preferring an *R*_1_-based technique. First, the concentration dependence is simpler to model compared with *R*_2_*, subject to some of the standard assumptions implicit in DCE-MRI analysis (e.g., fast water exchange regime). Second, it is more straightforward to measure *R*_1_ than *R*_2_* in blood vessels. However, the tissue *R*_1_ changes observed in this study were subtle and had wider distributions compared with corresponding *R*_2_* changes. This is partly due to the combined intravascular and extravascular effects of blood-pool contrast in *R*_2_* sequences, which leads to greater sensitivity in fMRI and perfusion imaging applications, and the superparamagetic nature of USPIO particles. A second challenge is the optimisation of *R*_1_ mapping for the wide range of values observed following USPIO administration. In future studies, it may be expedient to increase the precision of *R*_1_ measurement (for example by sacrificing spatial resolution) or to focus on *R*_2_* measurement where time is limited. For the latter approach, it is nevertheless useful to obtain *R*_1_ in blood for normalisation, which could be achieved more rapidly using a single-slice technique in place of a whole-brain measurement protocol [22]. A caveat is that while parenchymal uptake is expected to increase *R*_1_, the effect on *R*_2_* is more difficult to determine due to a more complex dependence on concentration, distribution and geometry [26,27]. Future studies should also assess CBV and parenchymal uptake in relation to small vessel disease severity, including WMH and enlarged perivascular space burdens. In conclusion, our work advances the methodology for probing the neurovasculature in SVD and for quantitatively assessing uptake of USPIO, paving the way for further use of these contrast agents in cerebral small vessel disease research and in other pathologies.

## 4. Materials and Methods

### 4.1. Participants

Twelve patients were recruited with a history of non-disabling minor ischaemic stroke occurring a minimum of one month previously. Patients were recruited from the regional clinical stroke service as described previously [6,28], and from our register of patients with a clinical diagnosis of minor non-disabling ischaemic stroke in the past five years. “Non-disabling” was defined as not requiring assistance in activities of daily living (modified Rankin score ≤ 3). We included those with diabetes mellitus, hypertension and other vascular risk factors as long as these were well-controlled. We excluded patients with unstable diabetes, chronic kidney disease with estimated glomerular filtration rate < 30, other neurological disorders, or other life threatening medical conditions. We also excluded patients unable to give consent, with contraindications to MRI, and who had haemorrhagic stroke (but not haemorrhagic transformation of an infarct). The study was conducted following Research Ethics Committee approval (South East Scotland Research Ethics Committee 01, reference 14/SS/1081, 27 November 2014) and according to the principles expressed in the Declaration of Helsinki. All subjects gave written informed consent.

### 4.2. MRI and USPIO Administration

#### 4.2.1. Patient Study

Participants were scanned using a 3-T Siemens Magnetom Verio MRI scanner with a 12-channel receive-only head coil (Siemens Healthcare, Erlangen, Germany). Scans took place at baseline (“scan 1_pre_”), which was immediately followed by USPIO (ferumoxytol; 4.0 mg/kg) infusion, at 24–30 h following infusion (“scan 2”) and at four weeks following infusion (“scan 3”). Eight of the patients were scanned additionally immediately after 1/3 of the total USPIO dose had been administered (“scan 1_post_”), with the remainder of the dose administered immediately after the scan. USPIO was administered outside of the scanner and was made up using a vial of ferumoxytol (30 mg/mL) diluted in up to 100 mL of sterile 0.9% sodium chloride to achieve a final concentration for infusion of 2–8 mg/mL, as recommended by the manufacturer. All participants received a final dose of 4.0 mg/kg, infused via a venous cannula sited in the participant’s forearm over at least 15 min with participants in a reclined or semi-recumbent position. A saline flush was administered following the infusion. Blood pressure was monitored and a physician was present at the time of administration and for at least 30 min afterwards to monitor the patient and treat any reactions.

Structural imaging consisted of *T*_1_w, *T*_2_w, GRE and *T*_2_-weighted FLAIR acquisitions. *R*_1_ and flip angle were measured using the DESPOT1-HIFI method [12] at scans 1_pre_, 1_post_ and 2, consisting of two 3D inversion-recovery prepared spoiled gradient echo sequences (IR-sGRE) and three 3D spoiled gradient echo sequences (sGRE) with different flip angles. *R*_2_* was measured using a 3D multi-echo spoiled gradient echo sequence (ME-sGRE) at all four scans. MRI sequence parameters are provided in Table 5.

#### 4.2.2. Phantom Validation

To verify the accuracy of the DESPOT1-HIFI protocol for a range of *R*_1_ values, we scanned a multi-compartment custom MRI phantom with a range of *R*_1_ [29]. This phantom consisted of nine sealed 10 mL syringes containing 0.05, 0.07, … and 0.21 mM MnCl_2_ in distilled water. The phantom container was filled with 1.5 g/L CuSO_4_ and 3.6 g/L NaCl solution to increase the coil loading, reduce susceptibility artefacts around the syringes and to provide a compartment with short *T*_1_ to mimic longitudinal relaxation in blood vessels following contrast injection [29]. Gold-standard measurements were obtained using a single-slice inversion-recovery spin-echo pulse sequence (TR/TE/TI = 1550/11/[30, 330, 730, 1130, 1530] ms). Since there is no accepted reference method for measuring *R*_2_* and the multi-echo spoiled gradient echo sequence is regarded as the standard method, we did not evaluate this sequence in vitro; however, phantom experiments can be used to evaluate repeatability, reproducibility and relaxivity properties [30].

### 4.3. Image Processing and Analysis

#### 4.3.1. *R*_1_ Mapping

To generate *R*_1_ parametric maps (Figure 1a), the signals of the sGRE and IR-sGRE scans were corrected for intensity scaling imposed by the scanner. To account for motion between the scans, all were co-registered to the first volume using rigid body registration (FSL FLIRT [31]). Reference [12] recommends modelling of the IR-sGRE signal under the assumption that the readout pulse trains do not influence the recovery of longitudinal magnetisation; for IR-sGRE with perfect inversion, this assumption leads to the following approximation for the steady-state signal:(1)$$S_{\mathrm{IR}-\mathrm{sGRE}}=S_{0}\frac{1-2e^{-R_{1}TI_{\mathrm{eff}}}+e^{-R_{1}TR}}{1+e^{-R_{1}TR}}\sin\kappa \beta$$
where *s*_0_ represents the maximum possible signal, *TI*_eff_ is the effective inversion time (i.e., the time between the inversion pulse and the excitation pulse for acquisition of the central *k*-space line), β is the flip angle corresponding to pulses in the readout train and *k* is the ratio of actual and nominal flip angles. To test the validity of this approximation, phantom data was modelled using both Equation (1) above and using the signal model expressed by Equation (13) in the article by Deichmann et al. [15], which does not make this assumption and which very closely approximates the complete pulse-by-pulse description derived by Brix et al. [32] The following additional characteristics of the IR-sGRE sequence implementation were taken into consideration: (i) the inversion pulse is non-selective and *B*_1_-insensitive, therefore complete inversion was assumed, and (ii) all *k*-space lines in the slice direction were sampled linearly in a single pulse train, thus the signal magnitude was calculated at the midpoint of the pulse train. Steady-state sGRE signals were modelled using the following well-known equation:(2)$$S_{\mathrm{sGRE}}=S_{0}\frac{1-e^{-R_{1}TR}}{1-e^{-R_{1}TR}\cos\kappa \beta }\sin\kappa \beta$$
where *T*_2_* decay is assumed to be negligible at the short echo times used. $s_{0}$, *R*_1_ and *k* were determined using in-house Matlab code (https://github.com/mjt320/HIFI) incorporating the Matlab lsqcurvefit function (MathWorks, Natick, MA, USA) to minimise the sum-of-square differences between the measured and predicted signals for each voxel. All three parameters were constrained to be positive. Additionally, *R*_1_ maps were obtained by fitting the sGRE phantom images alone with *k* fixed as unity in order to assess the accuracy of the widely-used variable flip angle technique. 

#### 4.3.2. *R*_2_* Mapping

To generate *R*_2_* maps (Figure 1b), multi-echo data was fit as described above using in-house Matlab code (https://github.com/mjt320/T2Star) but without constraints and using the signal model $s_{i}=s_{TE=0}e^{-R_{2}^{*}TE_{i}}$, where *s_i_* and *TE_i_* are the signal and echo time for echo *i,* respectively, and *s_TE=_*_0_ is the steady-state signal at zero echo time. To avoid fitting low SNR voxels and to reduce bias due to the noise “floor” at long TE, voxels and individual data points with intensities close to the noise level were excluded from the fitting using empirically determined thresholds.

#### 4.3.3. Structural Image Processing

Structural MR images were linearly co-registered to the gradient echo (GRE) image obtained at scan 1_pre_ using a two-step rigid-body registration (FSL FLIRT [31]). The brain was extracted automatically using FSL BET2 [33] with an empirically-determined weighted average of the *T*_2_-weighted (*T*_2_w) and GRE images as input, after correcting both images for the effects of coil sensitivity inhomogeneities and linearly normalising their intensities [34]. The intracranial volume mask was refined by nullifying voxels with standard intensity scores below −1.2 in the GRE image and filling any “holes” in the resultant mask. 

GM regions of interest (ROIs) representative of the deep grey matter were manually drawn as circular regions within the left and right thalamus on slices containing these structures, carefully avoiding tissue boundaries, lesions and vessels.

WM and WMH segmentation masks were obtained automatically using multispectral data fusion and a two-step Gaussian clustering followed by the computation of the Bayesian conditional probability for each cluster [35]. The first clusterisation operated on a four-dimensional array (three spatial dimensions plus sequence) formed by the combination of the brain-extracted *T*_1_w, *T*_2_w and FLAIR images, and generated likelihood maps of pure cerebrospinal fluid (CSF), normal-appearing white matter and a mixed tissue class containing features with high *T*_2_w and low *T*_1_w signal, including GM, WMH, SL and perivascular spaces. Deep GM voxels were removed from this map using a pipeline [36] that employs FMRIB Software Library tools and a cohort-relevant template; cortical GM was also removed via non-linear registration (NiftyReg, https://sourceforge.net/projects/niftyreg/) of a white matter template (http://datashare.is.ed.ac.uk/handle/10283) to subject space. A second clusterisation using GRE and FLAIR images was used to extract the FLAIR hyperintensities (i.e., SL and WMH) from any remaining elements of the mixed tissue class. Manual editing was then used to separate SL from WMH, guided by radiological reports. Likelihood maps were thresholded empirically to generate binary masks containing pure tissues with minimal contamination and were eroded to further reduce partial volume effects and the effects of any misregistration. Example masks and ROIs are shown in Figure 6.

#### 4.3.4. Quantitative Image Processing

*R*_1_ and *R*_2_* maps were spatially co-registered to the first time point using NiftyReg (non-linear transformation) and using the first image acquired for *R*_1_ and the first echo image of the *R*_2_* scan. To obtain *R*_1_ values in blood, regions were manually drawn within the superior sagittal sinus on three neighbouring slices, located centrally within the slab. Other ROIs were transformed into the parameter map space and used to sample *R*_1_ and *R*_2_* values. Median values for each patient and ROI were used in further analyses.

To account for inter-patient and inter-visit variation in blood-pool USPIO concentration, the normalised change in *R*_1_ was calculated at each post-USPIO time point as ∆*R*_1__,norm_ = ∆*R*_1_/∆*R*_1__,blood_, where Δ indicates the change relative to baseline (i.e., scan 1_pre_). Assuming a linear relationship between *R*_1_ and USPIO concentration, and assuming that USPIO is restricted to the blood pool immediately after infusion, this quantity approximates the cerebral blood volume fraction (*CBV*).

The normalised *R*_2_* change was also calculated. However, *R*_2_* values and changes in the superior sagittal sinus were spatially heterogeneous and inconsistent across patients, presumably due to the possibly complex [37], and to our knowledge, unknown dependence of *R*_2_* on venous USPIO concentration, vessel morphology, vessel orientation and location within the vessel [38]; therefore, blood *R*_1_ values were used for normalisation and the ratio $\Delta R_{2,\mathrm{norm}}^{*}$ = $\Delta R_{2}^{*}$/∆*R*_1__,blood_ was calculated, which is expected to be approximately proportional to the *CBV* immediately after infusion.

### 4.4. Statistics

Differences in relaxation rate between different time points or tissues were assessed using the two-sided paired *t*-test. Associations between *R*_1_ and *R*_2_* changes were explored using linear regression and Pearson’s correlation coefficient. A significance threshold of *p* < 0.05 was used.

## Figures and Tables

**Figure 1 ijms-20-00776-f001:**
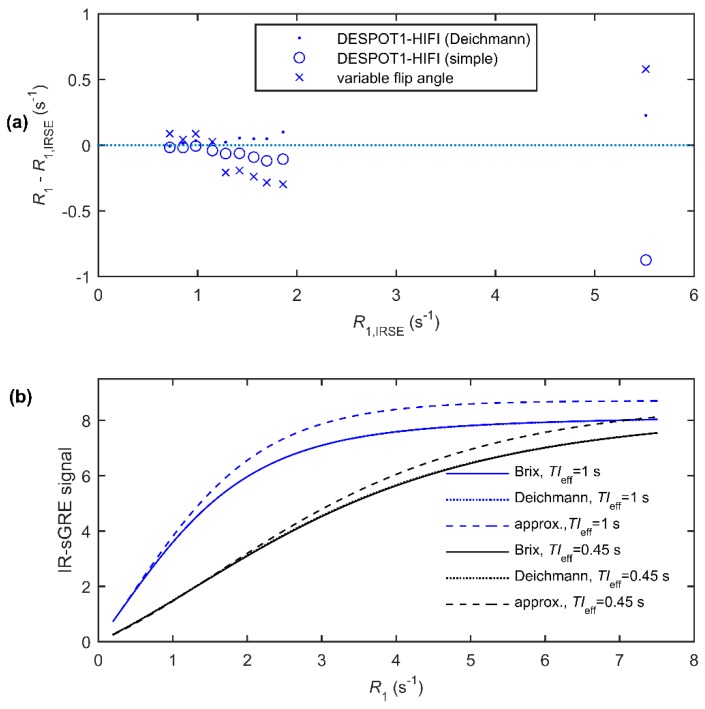
(**a**) Errors in the *R*_1_ measurements versus gold-standard inversion-recovery spin-echo (IRSE) values, measured in a phantom using DESPOT1-HIFI with the Deichmann signal model, an approximate signal model neglecting effects of the pulse train and using the variable flip angle method; (**b**) Calculated inversion-recovery prepared spoiled gradient echo (IR-sGRE) signal (assuming *s*_0_ = 100) for the acquisition parameters given in Table 5 using both the full mathematical description of Brix et al., the model by Deichmann et al. and the simplified model as described in the text. Note that the Deichmann prediction is almost identical to that of the Brix model and is consequently not visible.

**Figure 2 ijms-20-00776-f002:**
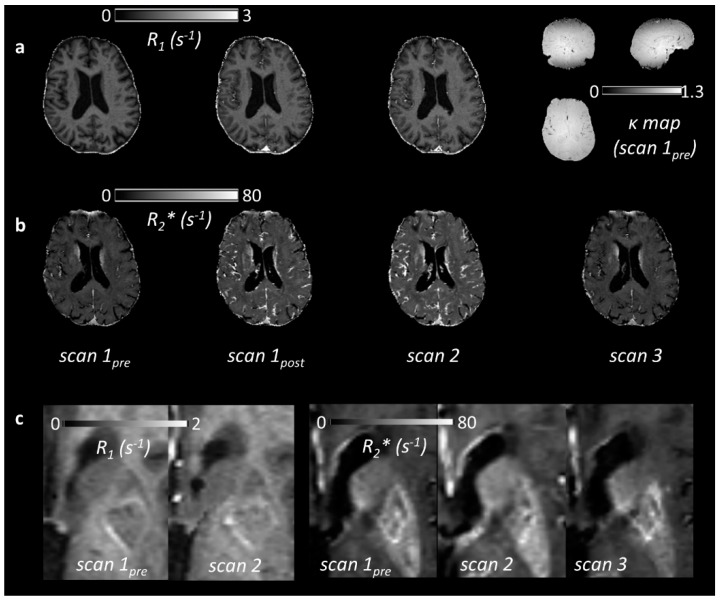
(**a**) *R*_1_ maps for a single patient at three time points: pre-ultrasmall superparamagnetic particles of iron oxide (USPIO) (scan 1_pre_), immediately post-USPIO (scan 1_post_), and at 24–30 h post-USPIO (scan 2). The corresponding relative flip angle (i.e., *k*: the actual flip angle divided by the nominal value) map measured at baseline is shown on the right, illustrating both variation due to *B*_1_ inhomogeneity and the (axial) slab excitation profile; (**b**) *R*_2_* maps in the same patient, obtained at the same time points and at 1 month post-USPIO (scan 3); (**c**) Parametric maps in another patient (79 year old male, first visit 31 days post-infarct in the left basal ganglia). *R*_1_ is seen to increase visibly in the stroke lesion following USPIO administration. The corresponding change to *R*_2_* is more subtle in the stroke lesion, but a visible increase is seen in the neighbouring background tissue.

**Figure 3 ijms-20-00776-f003:**
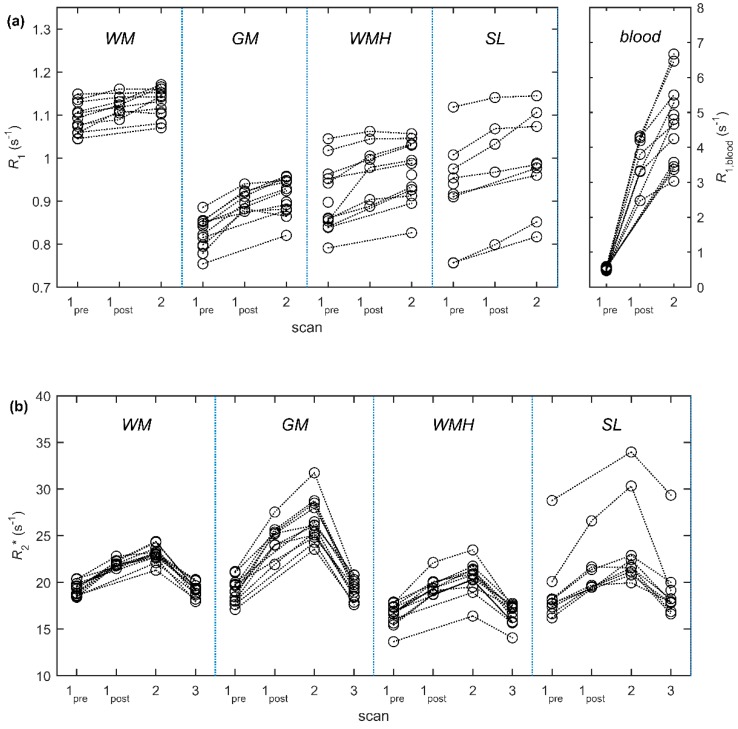
Median relaxation rates (**a**) *R*_1_ and (**b**) *R*_2_* for each participant and tissue region at scan 1_pre_ (pre-USPIO), scan 1_post_ (immediately post-USPIO), scan 2 (24–30 h post-USPIO) and scan 3 (1-month post-USPIO, *R*_2_* only). Abbreviations: WM: normal-appearing white matter, GM: grey matter, WMH: white matter hyperintensities, SL: stroke lesion.

**Figure 4 ijms-20-00776-f004:**
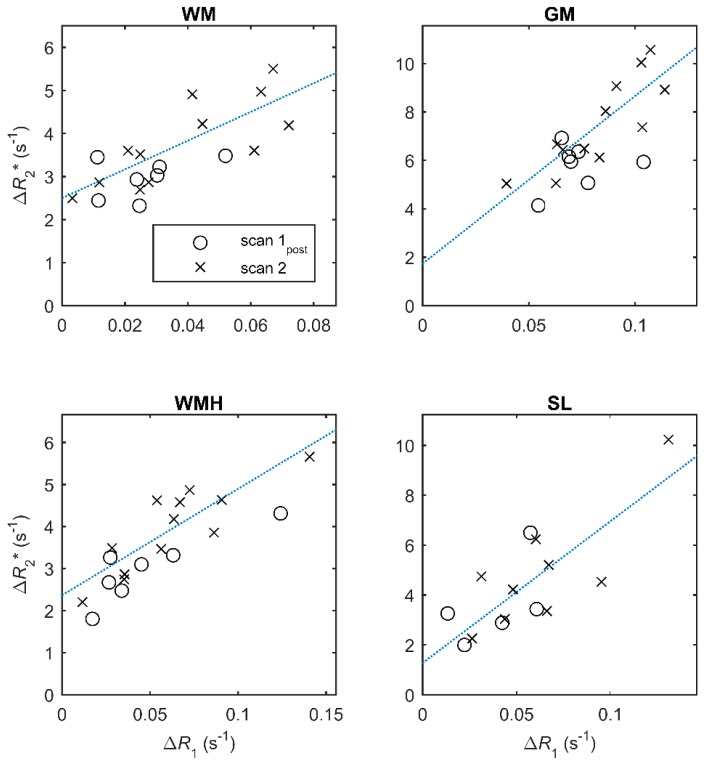
Change in *R*_2_* (Δ*R*_2_*) versus change in *R*_1_ (Δ*R*_1_) at scan 1_post_ (immediately post-USPIO; circles) and at scan 2 (24–30 h post-USPIO; crosses). Each data point represents the change in relaxation rate for a subject relative to the value measured at scan 1 (pre-USPIO). Dotted lines show the best fit to the data at scan 2; full linear regression results are given in Table 3.

**Figure 5 ijms-20-00776-f005:**
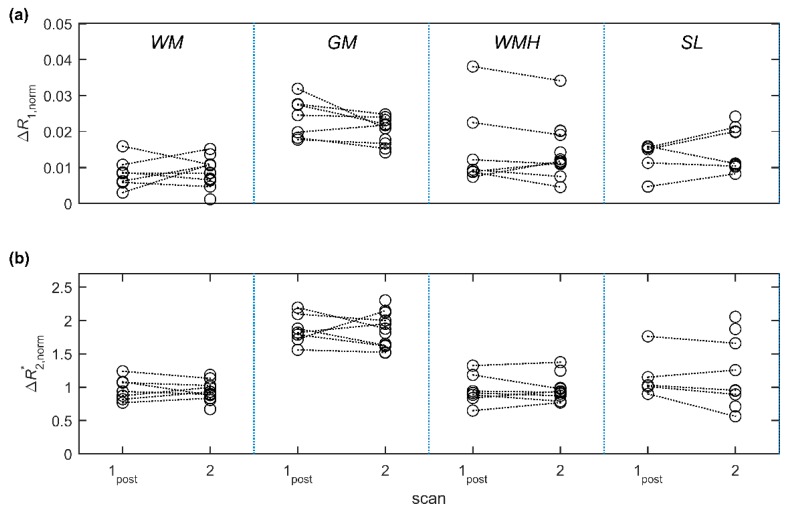
Blood-normalised relaxation rate changes (**a**) ∆*R*_1__,norm_ and (**b**) $\Delta R_{2,\mathrm{norm}}^{*}$ at scan 1_post_ (immediately post-USPIO) and scan 2 (24–30 h post-USPIO). Changes are relative to baseline values measured at scan 1_pre_ (pre-USPIO) and are normalised to corresponding *R*_1_ changes in blood. Values at scan 1_post_ are approximately equivalent (∆*R*_1__,norm_) or proportional ($\Delta R_{2,\mathrm{norm}}^{*}$ ) to cerebral blood volume fraction, while values at scan 2 are potentially influenced by additional parenchymal USPIO uptake. Abbreviations: WM: normal-appearing white matter, GM: grey matter, WMH: white matter hyperintensities, SL: stroke lesion.

**Figure 6 ijms-20-00776-f006:**
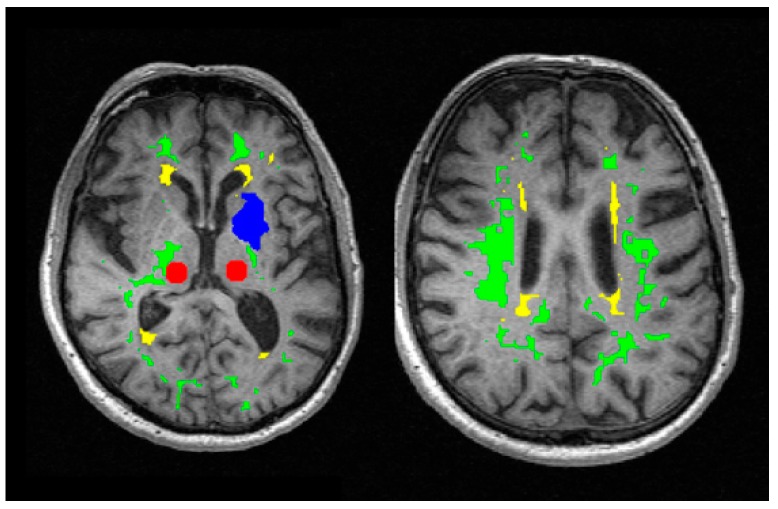
Regions of interest overlaid on the first image of the DESPOT1-HIFI series prior to USPIO infusion. Regions and segmentation masks shown are the manually drawn thalamus ROI (red), normal-appearing white matter (green), white matter hyperintensities (yellow) and the manually drawn stroke lesion ROI (blue).

**Table 1 ijms-20-00776-t001:** Patient demographics.

Characteristic	(*N* = 12)
Age (mean ± SD)	74.1 ± 6.7
Male	67% (8)
Prior Stroke or TIA	17% (2)
Diabetes	17% (2)
Hypertension	75% (9)
Hyperlipidaemia	58% (7)
Ischaemic Heart Disease	8% (1)
History of Smoking	50% (6)
Alcohol Intake (units per week, median and range)	9 (0–30)
NIHSS (median and range)	2 (1–4)
Modified Rankin Score Post Stroke (median and range)	1 (0–2)
Days Post-Stroke (median and range)	48 (31–276)
WMH Volume (mL) (median and range)	12.5 (3.2–46.9)

TIA: transient ischaemic attack; NIHSS: National Institutes of Health Stroke Scale; WMH: white matter hyperintensity.

**Table 2 ijms-20-00776-t002:** Mean (standard deviation) relaxation rate values for each region of interest (ROI), averaged over patients. Statistically significant changes relative to baseline are indicated by * (*p* < 0.05), ** (*p* < 0.01) and *** (*p* < 0.001).

ROI	*R*_1_ (s^−1^)			*R*_2_* (s^−1^)			
	Scan 1_pre_	Scan 1_post_	Scan 2	Scan 1_pre_	Scan 1_post_	Scan 2	Scan 3
blood	0.526 (0.036)	3.672 (0.629) ***	4.665 (1.140) ***	-	-	-	-
WM	1.090 (0.034)	1.123 (0.022) **	1.129 (0.032) ***	19.3 (0.7)	22.1 (0.4) ***	23.0 (0.8) ***	19.3 (0.7)
GM	0.825 (0.036)	0.907 (0.022) ***	0.908 (0.041) ***	19.1 (1.3)	24.9 (1.5) ***	26.6 (2.2) ***	19.3 (1.0)
WMH	0.905 (0.075)	0.983 (0.061) *	0.967 (0.067) ***	16.5 (1.1)	19.7 (1.0) ***	20.5 (1.6) ***	16.6 (1.1)
SL	0.926 (0.108)	1.002 (0.116) *	0.989 (0.102) ***	19.0 (3.6)	21.4 (2.5) **	23.9 (4.6) ***	19.3 (3.7)

WM: (normal-appearing) white matter, GM: grey matter, WMH: white matter hyperintensities, SL: stroke lesion.

**Table 3 ijms-20-00776-t003:** Linear regression analysis of change in *R*_1_ (Δ*R*_1_) versus change in *R*_2_* (Δ*R*_2_*) at scan 1_post_ (immediately post-USPIO) and scan 2 (24–30 h post-USPIO). Changes are calculated relative to scan 1_pre_ (pre-USPIO) values. Statistically significant associations are indicated by * (*p* < 0.05), ** (*p* < 0.01) and *** (*p* < 0.001).

ROI	Scan 1_post_			Scan 2		
	β_0_ (s^−1^)	β_1_	*R* ^2^	β_0_ (s^−1^)	β_1_	*R* ^2^
WM	2.6	14.2	0.18	2.5	33.4	0.61 **
GM	4.7	14.9	0.06	1.7	69.2	0.70 ***
WMH	2.1	18.7	0.75 *	2.4	25.3	0.73 ***
SL	1.8	47.2	0.34	1.3	56.9	0.64 **

β_0_: intercept, β_1_: slope, WM: normal-appearing white matter, GM: grey matter, WMH: white matter hyperintensities, SL: stroke lesion.

**Table 4 ijms-20-00776-t004:** Normalised relaxation rate changes for each ROI, averaged over patients. Coefficients of variation (%) are shown in parenthesis. Statistically significant differences compared with WM are indicated by * (*p* < 0.05), ** (*p* < 0.01) and *** (*p* < 0.001).

ROI	∆*R*_1__,norm_		$\Delta R_{2,\mathrm{norm}}^{*}$	
	Scan 1_post_	Scan 2	Scan 1_post_	Scan 2
WM	0.0084 (45)	0.0087 (43)	0.97 (16)	0.93 (14)
GM	0.0239 (21) ***	0.0204 (16) ***	1.86 (11) ***	1.85 (14) ***
WMH	0.0153 (68)	0.0148 (50) *	0.96 (22)	0.97 (17)
SL	0.0125 (34)	0.0151 (38) *	1.17 (26)	1.21 (41)

WM: normal-appearing white matter, GM: grey matter, WMH: white matter hyperintensities, SL: stroke lesion.

**Table 5 ijms-20-00776-t005:** MRI acquisition parameters.

Scan	Sequence	TR (ms)	TE (ms)	FA (°)	TI (ms)	FOV (mm)	Acquisition Matrix	Slices × Thickness (mm)	GRAPPA Factor	Time (m:ss)	Other
FLAIR	Axial 2D PROPELLER	9100	125	130	-	240	256	48 × 3	-	4:53	-
ME-sGRE	Axial 3D multi-echo spoiled gradient echo	50	4.6 8.5 14.0 19.5 25.0 30.5 36.0 41.5	15	-	240 × 240	256 (AP) × 192 (LR)	72 × 2	2	7:08	11.1% slice oversampling
DESPOT1-HIFI	Axial 3D IR-sGRE	1190	2.3	5	1000	240 × 240	256 (AP) × 192 (LR)	72 × 2	-	3:50	11.1% slice oversampling; echo-spacing = 4.5 ms for IR-sGRE
	Axial 3D IR-sGRE ^1^	632	2.3	5	450					2:03	
	Axial 3D sGRE	5.7	2.5	12	-					1:29	
	Axial 3D sGRE	5.7	2.5	5	-					1:29	
	Axial 3D sGRE	5.7	2.5	3	-					1:29	
GRE	Axial 3D gradient echo	40	20	15	-	240 × 240	320 (AP) × 256 (LR)	48 × 3	2	5:35	25.0% slice oversampling
*T*_1_w	Sagittal IR-sGRE	2300	2.98	9	1100	256 × 256	256 × 256	208 × 1	2	5:21	23.1% slice oversampling
*T*_2_w	Axial 2D PROPELLER	11,400	120	90	-	240	384	48 × 3	-	4:24	-

^1^ TR/TI = 832/650 ms for subjects 1 to 6; TR: repetition time; TE: echo time; FA: flip angle; TI: inversion time; FOV: field of view; AP: anterior/posterior; LR: left/right.

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
