# Peer review of "MRI Relaxometry for Quantitative Analysis of USPIO Uptake in Cerebral Small Vessel Disease"

_ijms, 2019, doi:10.3390/ijms20030776_

Round 1
Reviewer 1 Report
I find the relevance to the topic of the special issue ("Cellular and Molecular Mechanisms of Blood–Brain Barrier Dysfunction") quite cursory. Apart from this, the work is of interest, as it provides useful background information for people using USPIO-enhanced MRI to probe neuroinflammatory phenomena, is methodologically sound, and clearly written.
I have only a few minor comments:
· Abstract, line 32: “Our data are consistent with intravascular uptake and yield estimates of cerebral blood volume” I assume here the authors mean that parenchymal relaxation rate changes following USPIO administration were explained by blood volume alone, and failed to demonstrate parenchymal uptake by BBB disruption or intracellular. Please rephrase.
· Last sentence of the abstract should probably be “…The approach should be valuable in future studies to quantify both blood-pool USPIO …” as the focus here is on quantitation, rather than on sensitivity.
· At the end of the Introduction, when summarizing the work carried out, the authors should make clear that at 1 month post-injection only R2* maps were obtained (possibly explaining the reason why R1 maps were not available at this later time point). In addition, it should be specified here that blood R1 was measured in the superior sagittal sinus, briefly mentioning that blood R2* was not used due to its unreliability.
· Page 4, lines 9-10, “for R2*, a significant further increase was seen in WM, GM, WMH (p < 0.001) and in stroke lesions (p < 0.05).” I assume the authors refer to the difference between scan 1post and scan2 . Please specify
· The phantom used for checking the accuracy of R1 measures should be briefly described in the manuscript.
· Only the accuracy of R1 measures was verified. The rationale for not verifying accuracy of R2* measures should be discussed, or a similar validation should be carried out (e.g. as in MAGMA 2017;30:15-27).
· The authors mention that one patient was outlier at scan 1 post. Any hypothesis on the reason? How did this patient behave at scan 2 post?
Author Response
I find the relevance to the topic of the special issue ("Cellular and Molecular Mechanisms of Blood–Brain Barrier Dysfunction") quite cursory. Apart from this, the work is of interest, as it provides useful background information for people using USPIO-enhanced MRI to probe neuroinflammatory phenomena, is methodologically sound, and clearly written.
Thank you for the constructive feedback on our manuscript.
I have only a few minor comments:
· Abstract, line 32: “Our data are consistent with intravascular uptake and yield estimates of cerebral blood volume” I assume here the authors mean that parenchymal relaxation rate changes following USPIO administration were explained by blood volume alone, and failed to demonstrate parenchymal uptake by BBB disruption or intracellular. Please rephrase.
Yes – we have rephrased this sentence.
· Last sentence of the abstract should probably be “…The approach should be valuable in future studies to quantify both blood-pool USPIO …” as the focus here is on quantitation, rather than on sensitivity.
Changed.
· At the end of the Introduction, when summarizing the work carried out, the authors should make clear that at 1 month post-injection only R2* maps were obtained (possibly explaining the reason why R1 maps were not available at this later time point).
This has been added.
In addition, it should be specified here that blood R1 was measured in the superior sagittal sinus, briefly mentioning that blood R2* was not used due to its unreliability.
This has been noted.
· Page 4, lines 9-10, “for R2*, a significant further increase was seen in WM, GM, WMH (p < 0.001) and in stroke lesions (p < 0.05).” I assume the authors refer to the difference between scan 1post and scan2 . Please specify
Yes. This has been clarified.
· The phantom used for checking the accuracy of R1 measures should be briefly described in the manuscript.
Details of the phantom have been added to the methods section
· Only the accuracy of R1 measures was verified. The rationale for not verifying accuracy of R2* measures should be discussed, or a similar validation should be carried out (e.g. as in MAGMA 2017;30:15-27).
The aim of our phantom measurements was to confirm that our implementation of DESPOT1-HIFI yielded unbiased R1 across a wide range of values versus gold-standard inversion-recovery measurements. For R2* measurement, the multi-echo spoiled GRE method is generally accepted as a standard technique and there is not an accepted gold-standard method. Although this was not our primary aim, as described in the above MAGMA reference, the repeatability or reproducibility of R1 and R2* could be assessed using a phantom. A note explaining the reason for not performing in-vitro R2* measurements has been added with the above reference (4.2.2), and we have noted additional steps taken to reduce the impact of noise (4.3.2).
· The authors mention that one patient was outlier at scan 1 post. Any hypothesis on the reason? How did this patient behave at scan 2 post?
Despite much head scratching, there appears to be nothing unusual about this patient or scan. It may be that an operator or software error resulted in additional scanner adjustments during the T1 series, but that would be no more than a guess. We have noted that “no signs of protocol deviations or of unusual observations in relation to this participant were found in the study records or image header information.”

Reviewer 2 Report
The authors performed an interesting study on the investigation of USPIO uptake by MRI relaxometry in cerebral small vessel disease. The manuscript is well-written, has interesting findings, employs sound methods and the data are presented clearly. Furthermore, are careful discussion of the results is provided with reasonable interpretation which shows that the authors are aware of the limitations of this study. My main concern relates to the presentation and the explanation of the image postprocessing and image analysis methods which could be improved.
I made some comments below that the authors should take into consideration.
1. Introduction
Page 1, lines 42-44 and page 2, lines 1-2: The studies applying different imaging methodologies for investigating cerebral small vessel disease which are referred to in the text should be clearly cited, at least one example per methodology should be mentioned.
4. Materials and Methods
4.3.3. and 4.3.4.Structural and quantitative Image processing
-The way in which tissue segmentation and definition of ROIs were performed is not entirely clear to me. I believe that a slightly more detailed description accompanied by a figure could improve these paragraphs and ensure the repeatability of the current study by others.
-page 12, line 28: please explain what is meant by "other factors" which lead to the detected heterogeneity of R2* values in der superior sagittal sinus.
Statistical Analysis
As in addition to multiple t-tests, also linear regression analyses were conducted, this should be explained here as well.
Author Response
The authors performed an interesting study on the investigation of USPIO uptake by MRI relaxometry in cerebral small vessel disease. The manuscript is well-written, has interesting findings, employs sound methods and the data are presented clearly. Furthermore, are careful discussion of the results is provided with reasonable interpretation which shows that the authors are aware of the limitations of this study. My main concern relates to the presentation and the explanation of the image postprocessing and image analysis methods which could be improved.
Thank you for the helpful and constructive feedback on our manuscript. We have addressed the issue regarding presentation of the image processing as detailed below.
I made some comments below that the authors should take into consideration.
1. Introduction
Page 1, lines 42-44 and page 2, lines 1-2: The studies applying different imaging methodologies for investigating cerebral small vessel disease which are referred to in the text should be clearly cited, at least one example per methodology should be mentioned.
Relevant citations have now been added.
4. Materials and Methods
4.3.3. and 4.3.4.Structural and quantitative Image processing
-The way in which tissue segmentation and definition of ROIs were performed is not entirely clear to me. I believe that a slightly more detailed description accompanied by a figure could improve these paragraphs and ensure the repeatability of the current study by others.
The section describing structural image processing has been completely revised to improve clarity and an additional Figure 6 included showing example tissue masks and ROIs.
-page 12, line 28: please explain what is meant by "other factors" which lead to the detected heterogeneity of R2* values in der superior sagittal sinus.
The text has been revised and references added to note the orientation, concentration and geometric dependence of R2*, which (we presume) results in the heterogeneity.
Statistical Analysis
As in addition to multiple t-tests, also linear regression analyses were conducted, this should be explained here as well.
This information has been added.
